# Truck-Drone Delivery Optimization Based on Multi-Agent Reinforcement Learning

**Zhiliang Bi** [1], **Xiwang Guo** [1,*], **Jiacun Wang** [2], **Shujin Qin** [3] and **Guanjun Liu** [4]

1    School of Information and Control, Liaoning Petrochemical University, Fushun 113001, China; bizhiliang@stu.lnpu.edu.cn
2    School of Computer Science and Software Engineering, Monmouth University, West Long Branch, NJ 07764, USA; jwang@monmouth.edu
3    School of Economics and Management, Shangqiu Normal University, Shangqiu 476000, China; qinshujin@sqnu.edu.cn
4    School of Electronic and Information Engineering, Tongji University, Shangqhai 201804, China; liuguanjun@tongji.edu.cn
*    Correspondence: guoxiwang@lnpu.edu.cn

**Abstract:** In recent years, the adoption of truck–drone collaborative delivery has emerged as an innovative approach to enhance transportation efficiency and minimize the depletion of human resources. Such a model simultaneously addresses the endurance limitations of drones and the time wastage incurred during the "last-mile" deliveries by trucks. Trucks serve not only as a carrier platform for drones but also as storage hubs and energy sources for these unmanned aerial vehicles. Drawing from the distinctive attributes of truck–drone collaborative delivery, this research has created a multi-drone delivery environment utilizing the MPE library. Furthermore, a spectrum of optimization techniques has been employed to enhance the algorithm's efficacy within the truck–drone distribution system. Finally, a comparative analysis is conducted with other multi-agent reinforcement learning algorithms within the same environment, thus affirming the rationality of the problem formulation and highlighting the algorithm's superior performance.

**Keywords:** reinforcement learning; drone; multi-agent problem; path planning





## 1. Introduction

Truck–drone collaborative delivery is a delivery method that uses trucks as the carrier platform and drones as the main means of transportation. With the advancement of e-commerce and urbanization in recent years, coupled with technological progress and the escalating costs associated with human resource utilization, the global logistics industry has required a more efficient transportation model [1]. Unmanned aerial vehicles (UAV), or drones, known for their efficiency, speed, and ease of deployment, have garnered considerable attention. Consequently, the concept of truck–drone joint delivery has been introduced and received attention from the research society.

As shown in Figure 1, While there have been some algorithms and models that have explored the feasibility and time efficiency of truck–drone collaborative delivery, including initial tests and route design by companies such as FedEx, Meituan, and Amazon, most of them have primarily relied on heuristic algorithms for individual vehicles or treated the drones as a system parameter issue. There has been limited systematic exploration in this field, particularly in the context of truck–drone coordination, using reinforcement learning algorithms. With advancements in logistics network planning, drone positioning and control technology, the blueprint for truck–drone collaborative delivery projects in specific urban areas has become increasingly important [2,3].

Reinforcement learning has emerged as a focal point of research in recent years, not only in the field of control [4] but also in the domain of combinatorial optimization [5].

Reinforcement learning works by having an agent receive rewards based on its actions within an environment and subsequently evaluating its behavior patterns. Through iterative evaluations and adjustments of states and actions, a sequence based on Markov decision-making is developed to achieve the ultimate optimal objective. In this study, we employ reinforcement learning as the foundation to solve a truck–drone collaborative delivery problem under a multi-vehicle model. We have also made certain improvements to the algorithm. Furthermore, we compare the final model with different algorithms in various scenarios, thus validating the effectiveness of the algorithm and the model.

This study addresses the trajectory optimization problem for multiple agents in a fully cooperative game environment by partitioning the vehicle driving conditions and drone flight paths. We utilized multi-agent algorithms to achieve this objective. The number of drones and customers is among other factors as adjustable parameters within the environment. This approach effectively validates that, within a multi-agent environment, the partitioning of states and adjusting algorithm parameters can be universally applied to solve truck–drone delivery optimization problems.

The integration of reinforcement learning with the environment proves instrumental in effectively adjusting parameters such as the number of vehicles and customer service points [6]. By configuring different scenarios with varying vehicle and customer counts, we can simulate and validate the efficiency of the truck–drone delivery model across different residential density levels. Particularly in the context of multi-drone scenarios, adopting this fully cooperative mode based on game theory allows us to transform the multi-agent problem into a single-agent decision-making problem. This approach reduces extensive learning time and mitigates the challenge of dealing with a large joint action space.

Furthermore, the random exploration process introduced in this study yields optimal solutions more readily than traditional heuristic algorithms, eliminating the need for an additional parameter adjustment and encoding-decoding processes.

This paper presents a learning environment based on a real-world delivery model, considering multiple objectives and time constraints, atop the foundation of multi-truck and drone mixed delivery scenarios. We approach this problem from a reinforcement learning perspective, focusing on analyzing the flight trajectories of drones in the context of truck–drone delivery. The model encompasses attributes such as vehicle starting points, vehicle count, customer count, vehicle coordinates, and time. To address the challenge of a vast joint action space and potential conflicts or competition for rewards among agents introduced by reinforcement learning in multi-agent scenarios, we employ a multi-objective model. Building upon single-agent algorithms, we partition the vehicle routing problem into stages within a single simulation, effectively addressing issues inherent to traditional truck delivery models. The primary objective of this research is to minimize the drone flight time while maximizing customer service rewards, all while adhering to specified time constraints. Through reinforcement learning-based route optimization with different parameter settings within the same environment, we ultimately derive a set of potential optimal solutions.

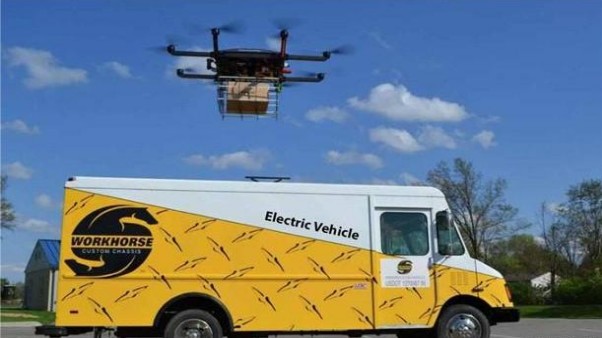

**Figure 1.** Truck–drone joint delivery.

The practical significance of this study lies in its expansion and optimization of delivery models and single-agent algorithm solutions for multi-agent cooperation. Traditional delivery models face challenges such as high labor costs, low efficiency, and the inability to deploy in urban environments rapidly. Examining the truck–drone delivery problem from the perspective of multi-agent algorithms reveals several key challenges. Firstly, adjusting the number of agents, i.e., vehicles and customers, is cumbersome and lacks flexibility. Secondly, there is the issue of defining joint action functions; in a multi-agent environment, actions of each agent mutually influence one another, leading to variations in reward functions and unstable states. Lastly, adopting truck–drone delivery models can effectively enhance delivery efficiency in urban areas, reducing time wastage and labor costs. This is especially pertinent in alleviating the labor intensity faced by logistics industry workers. In line with the multi-tiered site delivery systems currently popular in many countries' urban areas, this model reduces intermediary steps and minimizes the impact within the delivery process. Additionally, truck–drone transportation can benefit services requiring rapid transport capabilities, such as providing swift and precise services to shipping companies at ports through truck–drone deliveries [7,8]. In summary, the truck–drone delivery model represents a significant development direction for the future of the logistics industry and has the potential to become one of the representative modes of new-age logistics. It is poised to contribute to economic development and societal progress.

This paper makes the following specific contributions: (1) It investigates the truck–drone hybrid delivery problem using reinforcement learning, thereby deepening the research in the field of truck–drone delivery models. (2) It optimizes algorithms and environments for multi-agent scenarios, employing the Monte Carlo Tree Search as an optimization method. The approach divides the vehicle routing problem into stages, expediting the determination of truck routes, and employs multi-agent reinforcement learning algorithms for drone flight trajectories. Regularization techniques are utilized to enhance learning efficiency and stability. (3) Creates a versatile environment based on the truck–drone delivery model, allowing the adjustment of parameters such as the number of drones and customers. This environment offers a wealth of case studies for future research in truck–drone delivery.

*Related Work*

Logistics and distribution have long been a focal point of research in both academia and industry. In recent years, the growth of the e-commerce and food delivery industries has prompted numerous companies and research institutions to design and implement various new delivery models. Currently, most urban deliveries employ a multi-tiered station-based system, where deliveries are distributed through multiple levels of stations, starting from higher-level stations and progressing towards lower-level ones, until they encounter the "last-mile" problem. This results in significant waste of human resources and the creation of numerous intermediary steps. As is shown in Table 1, While there have been some studies and projects addressing the "last-mile" and logistics model modifications, it remains challenging to find a fundamental solution within traditional delivery models.

**Table 1.** Research direction.

| Research | [9] | [10] | [4] | [11] | This Study |
|---|---|---|---|---|---|
| Algorithm | Heuristic algorithm | MAPPO | MARL | AC&DQN | MAPPO |
| Model | MTSP | Nash equilibrium | Mean Field | FSTSP | Hierarchical |
| Optimization objective | Profit | Subsidiary communication | Multi-UAV systems | Delivery Time | Profit&Time |

With advancements in drone technology and related areas such as batteries, UAVs have observed widespread applications over the past decade across various industries [12], including the shipping sector [13,14], safety and security, parcel delivery, 3D modeling, disaster relief, and military applications. The drone delivery problem can be observed as an extension of routing problems and aircraft control domains. Compared to traditional delivery vehicles, drones offer high maneuverability, flight capabilities, and rapid

deployment, making them an attractive option in the delivery sector. Although they face challenges related to range and regulatory constraints, these issues have been mitigated to some extent through government regulations and the use of trucks as carrier platforms [15]. The truck–drone delivery model leverages the strengths of both trucks and drones. Trucks effectively address drone limitations in terms of range and payload capacity, while drones complement trucks in the final-mile delivery, especially in urban areas with high-rise buildings, reducing time and labor inefficiencies. This makes truck–drone delivery an appealing solution characterized by speed, cost-effectiveness, and adaptability [16].

Furthermore, truck–drone joint delivery entails different tasks and requirements compared to traditional vehicle delivery models. Notably, in terms of delivery routes, both trucks and drones must consider the delivery path concurrently. Additionally, the flight path of drones requires the consideration of numerous parameters, especially factors like obstacles and distances. When multiple drones are involved, even more variables come into play, and complex judgments are needed to account for the interactions between the flight trajectories of different quantities of drones. Beyond these considerations, the use of reinforcement learning algorithms aids in better decomposing the truck–drone delivery problem and allows researchers to conduct diverse testing, even in situations where real-world testing conditions are lacking. However, single-agent algorithms face limitations when dealing with action spaces and state spaces within multi-agent environments. Therefore, a more intelligent and generalized approach is needed to meet the demands of multi-drone flight trajectory and truck route planning in the context of truck–drone joint delivery. In summary, as a novel logistics delivery method, truck–drone hybrid delivery requires more experimentation and optimization to achieve commercialization and practical application.

Taking traditional heuristic algorithms as an example, recent years have observed significant research into the collaborative delivery of drones in the last mile delivery, such as the work by Júlia C. Freitas and their team on the truck–drone delivery problem [17]. This study introduced a novel Mixed-Integer Programming (MIP) formulation and a heuristic approach to tackle the problem. In all cases, the proposed MIP formulation yielded better linear relaxation bounds than previously proposed formulations and was capable of optimally solving several previously unresolved instances from the literature. The research employed a hybrid heuristic algorithm based on a generalized variable neighborhood search, combined with the concept of a taboo search, to obtain high-quality solutions for large instances. Extensive testing across numerous cases demonstrated the effectiveness of this approach.

In another study involving machine learning and truck–drone delivery by Yong Sik Chang and Hyun Jung Lee [18], the focus was on finding an efficient transport route for a truck carrying drones. After an initial K-means clustering and Traveling Salesman Problem (TSP) modeling, the research aimed to find offset weights for moving cluster centers, allowing for a wider drone delivery area along shorter truck routes. To validate the effectiveness of the proposed model with transfer weights, it was compared with two other delivery route methods. Experimental results using paired t-tests on randomly generated delivery locations demonstrated that the proposed model outperformed the other two models.

The study [19,20] combined traditional problems and addressed the TSP with a single drone and a truck. Mohammad Moshref-Javadi and his team assumed that the truck could stop at customer locations and release the drone multiple times in a stopped state for delivery. This extended the Traveling Repairman Problem (TRP), and a hybrid tabu search simulated annealing algorithm was used to solve the problem.

Under the consideration of policy and weather factors [21], HoYoung Jeong, Byung-Duk Song, and Seokcheon Lee accounted for the impact of package weight on drone energy consumption and flight-restricted areas. Particularly, the flight range of drones was highly affected by load weight, and drones were not allowed to fly over sensitive facilities regulated by the Federal Aviation Administration (FAA) in the United States, nor were they

allowed to fly in certain areas temporarily due to weather conditions. The study established a mathematical model incorporating these factors and proposed a two-stage constructive algorithm and search heuristic algorithm to enhance the computational efficiency for real-world case problems.

In the realm of reinforcement learning [11], Liu et al. extended the Flexible Service Technician Scheduling Problem (FSTSP) with stochastic travel times and formulated it as a Markov Decision Process (MDP). This model was solved using reinforcement learning algorithms, including Deep Q-Networks (DQN) and Advantage Actor-Critic (A2C) algorithms, to overcome the curse of dimensionality. Testing was conducted using artificially generated datasets, and the results demonstrated that, even in the absence of stochastic travel times, reinforcement learning algorithms outperformed approximate optimization algorithms and performed better than MIP models and local search heuristic algorithms on the original FSTSP.

The articles mentioned above largely employ traditional heuristic algorithms such as genetic algorithms to address optimization problems in the context of truck–drone joint delivery. However, this paper utilizes reinforcement learning and the MPE library to optimize the truck–drone delivery model in a multi-objective setting based on customer and vehicle parameters. In contrast to traditional algorithms, the approach proposed in this paper better considers real-world requirements such as vehicle scheduling and time coordination. Additionally, by introducing the multi-agent problem, adjustments can be made for different vehicle quantities, making the results closer to real-world scenarios.

Currently, route problems related to pure truck models with time as the primary metric are referred as MLP or TSP. This problem is commonly known as the Customer-Centric Routing Problem and is typically a single-objective problem aimed at minimizing time. Considering the unique advantages of drones in achieving faster deliveries, this paper emphasizes the multi-agent problem, which is addressed through reward mechanisms and constraint conditions to capture the complexities of real-world scenarios better [22].

The rest of the paper is organized as follows: Section 2 outlines the mathematical model and the environment of the problem. Section 3 introduces the reinforcement learning algorithms and key definitions. Section 4 conducts a comparative study of the algorithm's performance. Section 5 concludes the paper and presents future prospects.

## 2. Problem Description

In this section, we will outline the differences between the delivery model designed in this paper and traditional truck delivery systems. We will also provide a detailed explanation of the model's assumptions and clarify the distinctions between multi-agent and single-agent reinforcement learning in the context of the truck–drone environment.

Truck–drone delivery is a multi-tiered hybrid delivery model typically composed of multiple trucks, each equipped with a number of drones, as shown in Figure 2. In this model, drones are employed as delivery tools to replace traditional delivery personnel, while trucks play multiple roles as storage depots, command centers, maintenance hubs, launch platforms, and energy replenishment stations. In early drone delivery concepts, the primary idea was to replace traditional last-mile delivery personnel with drones, effectively reducing labor costs. However, due to variations in delivery time windows, significant drone idle time was incurred. Moreover, because of the limited maximum range of drones, they could only operate within a circular area centered on the delivery center with the maximum range as the radius. This inadvertently increased the cost of establishing a delivery network covering urban areas.

In this study, drones are conceptualized as being centered around the trucks, forming a circular delivery area with the maximum range as the radius. However, this circular area can move with the trucks, and the trucks themselves serve as multi-functional platforms to assist drones in delivery. This approach effectively enhances delivery efficiency and reduces the cost of building a logistics network.

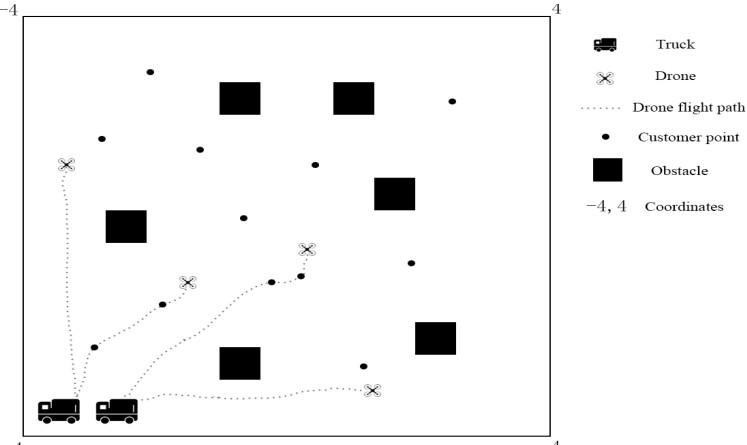

**Figure 2.** Simulated environment for a $8 \times 8$ square mile area.

In this study, the coordination of multiple trucks and drones is planned within a graph-based environment centered around customer service points, where the service points serve as nodes to provide services to customers. This approach is based on real-world urban residential environments and is advantageous for simplifying the mathematical model and the state space in reinforcement learning.

In this research, multiple trucks are simultaneously engaged in delivery, and the objective function involves multiple goals. Moreover, the intelligent agents in this model exhibit a fully cooperative relationship, akin to cooperative game theory. To enhance the efficiency of the reinforcement learning algorithm, the study promotes individual action selection through an overarching objective. The training of agents is conducted in stages, with the trajectory of the previous agent's training behavior serving as the learning environment for the next agent. This approach accelerates the learning efficiency of the reinforcement learning algorithm. Regarding drones, this study simplifies the drone flight model and abstracts the service points using existing drone route planning software and satellite imagery. This abstraction simplifies path parameters, particularly in considering factors such as wind conditions and urban obstacles.

To solve the truck–drone delivery problem in the multi-agent context, this research employs multi-agent deep reinforcement learning algorithms like Multi-Agent Proximal Policy Optimization (MAPPO) and Multi-Agent Deep Deterministic Policy Gradient (MADDPG). Utilizing these multi-agent algorithms, and adopting a centralized processing approach in a fully cooperative mode, the study successfully simulates and solves the truck–drone delivery problem using game theory and mathematical modeling.

### 2.1. Notations and Model

The section presents the optimization model and notations used in the model. This model incorporates elements from some other algorithmic research related to the TSP [23,24]. However, based on the algorithms and simulation scenarios presented in this paper, we have extended and modified some of the formulas and definitions to better align with the delivery problem in the context of truck–drone coordination.

#### 2.1.1. Mathematical Notations

1.  $\mathbb{P}$ The set of parking points $\mathbb{P} = \{1, 2, \ldots, p\}$.
2.  $\mathbb{C}$ The set of customers $\mathbb{C} = \{1, 2, \ldots, c\}$.
3.  $\mathbb{U}$ The set of drones $\mathbb{U} = \{1, 2, \ldots, u\}$.
4.  $\mathbb{V}$ The set of trucks $\mathbb{V} = \{1, 2, \ldots, v\}$.
5.  $t$ Unit time. It can be regarded as a step of the algorithm in this study.
6.  $v^t$ Truck speed.
7.  $v^d$ Drone flight speed.
8.  $\mathbb{C}_p$ The set of internal customer numbers at parking points $p$, where $p \in \mathbb{P}$.

9. $d_{i,j}^V$ The distance of the truck from parking point $i$ to parking point $j$, where $i, j \in \mathbb{P}$.
10. $d_{i,k}^U$ The distance of the truck from parking point $i$ to customer $k$, where $i \in \mathbb{P}, k \in \mathbb{C}$.
11. $r_{i,p}$ The reward received from customer $i$ at parking points $p$, where $i \in \mathbb{C}, p \in \mathbb{P}$.
12. $t_{i,j}^T = d_{i,j}^V / v^t$; the travel time of truck from point $i$ to point $j$, where $i, j \in \mathbb{P}$.
13. $t_{i,k}^D = 2 * d_{i,k}^U / v^d$; the flight time of the drone from parking point $i$ to customer $k$, where $i \in \mathbb{P}, k \in \mathbb{C}_p$.
14. $t_p^{av}$ Total time spent at parking point $p$, where $p \in \mathbb{P}$.
15. $P_i$ If parking point $i$ is visited, $P_i = 1$; otherwise, $P_i = 0$. $i \in \mathbb{P}$.
16. $C_k$ Total time accumulated before serving customer $k$, where $k \in \mathbb{C}$.
17. $R_p$ Total customer reward at parking point $p$, where $p \in \mathbb{P}$.

### 2.1.2. Decision Variables

1. $V_{i,j}^n$ If truck $n$ moves from node $i$ to node $j$, then $V_{i,j}^n$ is equal to 1, otherwise it is 0, $\forall n \in \mathbb{V}$
2. $U_{i,j}^n$ If drones $n$ moves from customer $i$ to customer $j$, then $U_{i,j}^n$ is equal to 1, otherwise it is 0, $\forall n \in \mathbb{U}$.

### 2.1.3. Objective Functions

$$min\ C_k \quad C_k = \sum_{i \in \mathbb{P}} \sum_{j \in \mathbb{P}} \left( t_{i,j}^T * V_{i,j}^n \right) + \sum_{i \in \mathbb{P}} P_i * \left( \sum_{k \in C_p} \sum_{i=1}^{k-1} t_{i,k}^D * U_{i,j}^n \right)$$

$$max\ R_p \quad R_p = \sum_{i \in \mathbb{P}} P_i * \sum_{i \in C_p} r_{i,p}$$

### 2.1.4. Constraints

(1) To avoid subsequent scheduling issues, the travel time gap between any two trucks should not exceed 30%.

$$0.7 \le \sum_{v \in \mathbb{V}} (t_{i,j}^{T_{v1}} / t_{i,j}^{T_{v2}}) \le 1.3$$

(2) The order reward will fade to a certain extent over time, but it cannot be less than 0.

$$r_{i+1,p} = r_{i,p} - f(C_k) \quad \forall r_{i,p} \ge 0 \quad f(C_k) = C_k - C_k(t_p^{av}/8T)$$

(3) Each drone can only be used by one truck at a time.

$$\sum_{n \in \mathbb{U}} \sum_{i \in \mathbb{C}} \sum_{j \in \mathbb{C}} U_{i,j}^n \le 1 \quad \forall i \in \mathbb{C} \quad \forall j \in \mathbb{C}$$

(4) Each parking point should be visited at least once by a truck.

$$\sum_{j \in \mathbb{P}} V_{i,j}^n = 1 \quad \forall n \in \mathbb{V} \quad \forall i \in \mathbb{P}$$

(5) Ensure consistency between truck and drone movements. If a truck is moving from parking point $i$ to parking point $j$, the drone should move from customer $i$ to customer $j$

$$V_{i,j}^n = U_{i',j'}^n \quad \forall n \in \mathbb{U} \quad \forall i, j, p \in \mathbb{P} \quad \forall i', j' \in \mathbb{C}_p$$

The objective function (1) is formulated with the aim of minimizing the delivery time based on the minimization of customer waiting time. The objective function (2) is associated with the intrinsic value of the customers' orders or potential rewards, which can be considered as tips or satisfaction ratings, and it decreases as a function related to waiting time, but never falls below zero. Together, these two objective functions constitute the optimization objectives of this study. In this study, based on the real-world model, it is stipulated that each customer must be visited by a drone, and each truck must stop at every delivery point to ensure that every customer receives the service they require. Constraint

(1) is included to address scheduling issues that may arise due to significant disparities in working hours among trucks, even in cases where multiple trucks are involved in delivery. Constraint (2) ensures that each delivery point is visited, thus guaranteeing service for every customer. Constraint (3) defines customer rewards, which are influenced by the order value and delivery time.

*2.2. Reinforcement Learning for Truck–Drone Delivery*

In this section, we provide a comprehensive overview of the algorithmic components and their application in our study to address the truck–drone delivery challenge. We emphasize the utilization of reinforcement learning techniques for route optimization within the framework of truck–drone coordination. Furthermore, we spotlight the incorporation of multi-agent algorithms for managing drone flight trajectories and elaborate on the integration of Monte Carlo Tree Search as an exploration strategy for optimizing truck routes. Additionally, we employ reward environment normalization techniques to enhance the efficiency and speed of the algorithm in attaining optimal solutions. Moreover, we conduct comparative assessments involving various algorithms and frameworks to validate the effectiveness of our model and reinforcement learning algorithms.

Reinforcement learning, inspired by behaviorist psychology, serves as a foundational learning approach aimed at refining an agent's behavior through interactions with its environment to maximize cumulative rewards. Within multi-agent settings, reinforcement learning can be categorized into cooperative and competitive scenarios, each dedicated to optimizing collective or competitive objectives across multiple agents. Research on multi-agent reinforcement learning extends to game-theoretic models that consider the intricate interactions between intelligent agents and employs deep reinforcement learning techniques, leveraging neural networks to enhance decision-making capabilities. These research directions seek to empower reinforcement learning systems to collaborate more effectively in complex environments and optimize their decisions to achieve higher rewards or accomplish more intricate objectives.

In our study, we introduced an additional element of random exploration through Monte Carlo Tree Search at the early stages of the reinforcement learning process, specifically in the context of vehicle routing environments. This augmentation serves to facilitate the rapid learning of the environment by intelligent agents while preventing them from becoming trapped in local optima.

The incorporation of neural networks in reinforcement learning, particularly deep reinforcement learning, provides a means to mitigate the "curse of dimensionality". In this investigation, deep reinforcement learning is selected as the primary focus of our testing efforts. Moreover, this research uses third-party libraries, including the Gym and the MPE library (built upon Gym), to allow for more comprehensive algorithm testing. The research environment is intentionally designed with adjustable parameters, enabling testing across various scenarios to showcase the effectiveness of both the model and the algorithms.

In the environment definition of this study, multiple intelligent agents make decisions in stages. After the first intelligent agent explores and learns within the environment, weight evaluations based on action trajectories are performed. This process incorporates the behavior of the first intelligent agent as part of the new environment. Ultimately, in a scenario of complete cooperative gameplay, centralized processing is used to address multi-agent problems. This approach effectively reduces the complexity of establishing joint action functions and significantly compresses the state space of multi-agents. It also reduces exploration and learning time, thereby greatly enhancing the feasibility of reinforcement learning for truck–drone delivery planning within urban areas.

In terms of the mathematical model, this study is based on the Vehicle Routing Problem (VRP) and adjusts it to accommodate vehicles and service points while introducing drones as new parameters. This research aims to make the mathematical model more closely aligned with real-world multi-vehicle logistics scenarios and enhance its versatility to meet the needs of various practical application areas. While focusing on solving the VRP,

the introduction of drones as an innovative parameter broadens the scope of applications for the research community. This model introduction holds significant academic and practical value for addressing complex supply chain, logistics, and transportation issues in the real world.

This paper first defines a multi-objective problem based on time constraints regarding the objective function. From a logistics perspective, both time and efficiency are optimization objectives of paramount importance. Additionally, as a replacement for multi-level distribution systems, we also need to consider customer satisfaction during last-mile deliveries. Therefore, we optimize vehicle travel time and customer rewards as the main objectives, aiming to improve efficiency at the macro level and service quality at the micro level. Finally, as providers of delivery services, we must serve each customer, necessitating that vehicles visit every stopping point, which forms the foundation of the TSP. Concerning vehicle coordination, we consider the need for balanced completion times among vehicles when serving urban areas, resulting in constraints on the equality of vehicle completion times. From the perspective of drones, we employ multi-agent reinforcement learning for drone coordination in delivery and introduce environmental parameters such as collisions and obstacles.

## 3. Principle and Application

### 3.1. Policy-Based Reinforcement Learning and Policy Gradient

This research primarily employs the MAPPO algorithm, which is a variant of deep reinforcement learning based on the AC framework. AC is a type of deep reinforcement learning algorithm initially proposed by Barto et al. [25]. In 1983, and it serves as the foundation for the popular Actor–Critic architecture used today. AC is a policy-based reinforcement learning method. In AC, the actor is responsible for interacting with the environment, while the critic acts as an observer and is responsible for updating the relevant value or policy networks. The update process involves the actor interacting with the environment, generating data, and calculating gradients. These gradients are then passed to the critic, which updates its network parameters using methods like SGD-based backpropagation. Once the parameters are updated, they are synchronized back to the corresponding actor for further interaction. AC algorithms offer several advantages in most testing scenarios, including better convergence properties and greater efficiency in high-dimensional and continuous action spaces. These advantages align well with the objectives and constraints of this research, facilitating the optimization problem-solving process.

The neural network structure within the AC algorithm framework also enhances memory and processing capabilities, helping to address the "curse of dimensionality" and improve the algorithm's versatility. Below, we will provide a brief introduction to the principles of the AC algorithm and discuss its potential applications and advantages in this research. Additionally, AC algorithms have various variants and initial versions, such as the A3C algorithm or multi-agent MA-A3C algorithms. Many other multi-agent algorithms also draw inspiration from the AC framework. Using this algorithm as a testbed contributes to future comparisons with other multi-agent algorithms.

The AC (Actor–Critic) framework is policy-based, and its early incarnation was the QAC algorithm, which originated from the policy gradient (PG) method. This framework serves as the prototype for most policy-based reinforcement learning algorithms. The Actor–Critic algorithm can be divided into two parts. The Actor component, which has its roots in policy gradient methods, can select appropriate actions in continuous action spaces, a capability that value-based Q-learning lacks, as it primarily addresses discrete action spaces. However, the Actor's learning efficiency is relatively slow because it updates based on the return of an entire episode. At this point, introducing a value-based algorithm as the Critic allows for single-step updates using TD methods, effectively trading bias for variance. This complementary relationship between the two algorithms forms the basis of the Actor–Critic framework. In the Actor–Critic framework, the Actor selects actions based on a probability distribution, the Critic evaluates the scores of actions generated by

the Actor, and the Actor subsequently modifies the probability of selecting actions based on the Critic's scores.

Like other reinforcement learning algorithms, AC algorithms utilize basic components such as time (t), states (s), rewards (r), and actions (a) to define their environment. In policy-based reinforcement learning and AC algorithms, the learning of policy parameters is often based on the gradient of some policy performance metric $J(\theta)$. Since the objective of these methods is to maximize this metric, their updates are akin to ascending the gradient of $J$.

$$\theta_{t+1} = \theta_t + \alpha \widehat{\nabla J(\theta_{t+1})}$$

In this context, $\widehat{\nabla J(\theta_{t+1})}$ represents an approximate estimate of the gradient of the performance metric parameter $\theta$.

The introduction of the state–value function allows AC algorithms to perform a secondary evaluation of action values. In the TD(0) version of the AC model, a differentiable parameterized policy $\pi(a|s, \theta)$ is used as the input along with a differentiable parameterized value function $v(s, w)$. The learning process involves iteratively updating the initial policy parameters $\theta$ and the weights w of the state–value function. For each non-terminal state $S$ in each subsequence, an action $A$ is taken. The transition to the next state $S'$ after taking action $A$ is observed, and both the weights $w$ and policy parameters $\theta$ are updated. This process allows for the evaluation of the value of state $S$.

In the case of continuous problems without subsequence boundaries, the policy is defined based on the average return at each time step $t$.

$$J(\theta) \doteq r(\pi) \doteq \lim_{h \to \infty} \frac{1}{h} \sum_{t=1}^{h} \mathbb{E}[R_t | S_0, A_{0:t-1} \sim \pi]$$

$$= \lim_{t \to \infty} \mathbb{E}[R_t | S_0, A_{0:t-1} \sim \pi]$$

$$= \sum_s \mu(s) \sum_a \pi(a|s) \sum_a p(s', r|s, a)r$$

where $\mu$ represents the stationary distribution under policy $\pi$, $\mu(s) \doteq \lim_{t \to \infty} Pr\{S_t = s | A_{0:t-1} \sim \pi\}$ and it is assumed to exist and be independent of $S_0$. Similar to problems with subsequence boundaries, in continuous problems, we use the differential return to define the value function.

$$\sum_s \mu(s) \sum_a \pi(a|s, \theta) p(s', r|s, a) = \theta(s') \qquad \forall s' \in S$$

At the same time, although in some environments where the sole value is the objective function, policy-based reinforcement learning may be inferior to value-based reinforcement learning in the final results due to the slow convergence of policy updates; policy parameterization has an important theoretical advantage over action–value-based methods. Policy parameterization provides a clear formula to show how the performance metric is influenced by policy parameters without involving derivatives of state distributions. This formula serves as a theoretical foundation for all policy gradient methods.

As shown in Figure 3. In the PG algorithm, the Agent is also referred to as the Actor. Each Actor has its own policy $\pi$ for a specific task, typically represented by a neural network with parameters $\theta$. Starting from a particular state and continuing until the end of the task, which is known as an episode, at each step, a reward $r$ is obtained; the total reward obtained for completing the task is denoted as R. In this way, an episode with T time steps involves the agent interacting with the environment, forming a sequence $\tau$, as follows:

$$\tau = (s_1, a_1, s_2, a_2, \ldots, s_T, a_T)$$

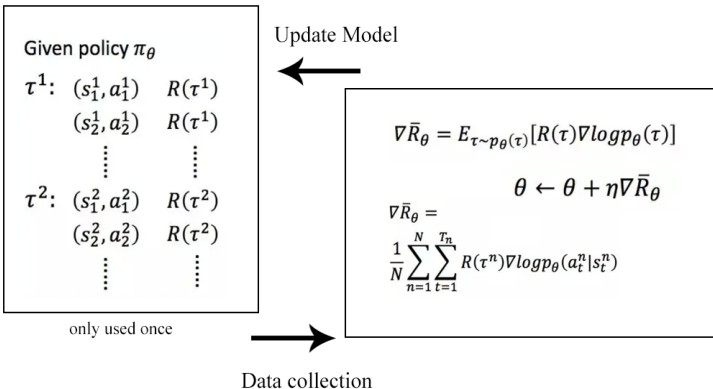

**Figure 3.** Policy Gradient.

The total reward obtained by the sequence $\tau$ is the sum of the rewards obtained at each stage, denoted as $R(\tau)$. Therefore, in the case of the Actor's policy being $\pi$, the expected reward that can be obtained is:

$$\overline{R}_\theta = \sum_\tau R(\tau)P_\theta(\tau) = E_{\tau \sim P_\theta(\tau)}[R(\tau)]$$

The objective of the algorithm is to adjust the Actor's policy $\pi$ to maximize the expected reward. This is achieved through the policy gradient method, which uses the gradient ascent to update the network parameters $\theta$ (i.e., update the policy $\pi$). Ultimately, the focus of the algorithm shifts towards finding the gradients of the parameters.

Under the AC framework, the Trust Region Policy Optimization (TRPO) algorithm was developed, based on a natural policy gradient optimization, and quickly gained popularity. While natural policy gradient optimization helps optimize policies in reinforcement learning algorithms, improving their performance and convergence speed, it still has certain drawbacks, such as not checking whether updates actually improve the policy [26].

Moreover, the TRPO algorithm itself has certain limitations. For instance, it can be challenging to determine how to improve performance when TRPO training does not yield the expected results. TRPO's actual implementation is constraint-based and involves computing the Fisher matrix, which significantly slows down the updating process.

*3.2. Multi-Agent Proximal Policy Optimization*

As a result, in 2017, John Schulman and others introduced the PPO algorithm, building upon the foundation of the TRPO algorithm [27]. In the PPO algorithm, it alternates between data sampling through interaction with the environment and optimizing the "agent" objective function using a stochastic gradient ascent. Unlike standard policy gradient methods that perform a single gradient update for each data sample, PPO introduces a new objective function that allows for multi-epoch minibatch updates. PPO inherits some advantages from TRPO but is much simpler to implement, more general, and exhibits better sample complexity in practice.

In the context of the problem and environmental characteristics of this study, the PPO algorithm offers several advantages over other reinforcement learning algorithms: (1) Stability through Clipping: PPO constrains policy updates within a smaller range by introducing clipping and probability ratio constraints. This helps prevent drastic fluctuations in gradient updates, enhancing training stability and reliability. (2) Compatibility with Various Networks: PPO is compatible with various neural network structures and function approximators, including deep neural networks. This makes it more versatile across different domains and tasks, allowing for easier adjustments of environmental and algorithm parameters in the context of truck–drone delivery research. (3) Adaptability to Continuous and High-Dimensional Spaces: PPO is well-suited for addressing problems in continuous

action spaces and high-dimensional state spaces. This adaptability is particularly valuable for solving issues related to drone flight trajectories.

These advantages make PPO an attractive choice for solving complex reinforcement learning problems. However, in the context of this research, which involves multiple agents in a delivery environment, the original PPO algorithm, although extendable, faces challenges in handling multi-agent coordination or competition. This is especially relevant when dealing with multiple drones operating in the same environment. Therefore, this study employs the MAPPO algorithm, explicitly designed for multi-agent environments where multiple agents need to make coordinated decisions. As shown in Algorithm 1, MAPPO introduces the concept of multiple agents to better handle cooperative decision-making and interactions among multiple intelligent entities.

---

**Algorithm 1** Multi-Agent PPO

---

Initialize critic $Q_{w^u}$ and actor $\pi^u$ with $\theta^u, \forall u$

Initialize the current policies $\pi^u_{old}$ with $\theta^u_{old} \leftarrow \theta_u$ , and the target critic $Q_{\overline{w}^u}$ with $\overline{w}^u \leftarrow w^u$

  Initialize a memory buffer $D$

  **for** iteration = $1, 2, \ldots, L$ **do**

   $s_1$ = initiate state

   **for** an episode $t = 1, 2, \ldots, T$ **do**

    Each agent $u$ takes action according to $\pi^u_{old}(a^u_t | z^u_t)$

    Get the reward $r_t$, and the next state $s_{t+1}$

   **end for**

   Obtain a trajectory for each UE $u : \tau^u = \{z^u_t, a^u_t, r_t\}^T_{t=1}$

   Compute $\{\hat{Q}^u(s_t, \mathbf{a}_t)\}$

   Compute advabtages $\{A^u(s_t, \mathbf{a}_t)\}$

   Into data buffer $D$

   **for** mini-batch $k = 1, \ldots, K$ **do**

    **for** each-policy $i = 1, \ldots, n$ **do**

     $b \leftarrow$ random mini batch from $D$ for policy $i$

     **for** each data chunk $c$ in the mini batch $b$ **do**

      Update the RNN hidden states for each timestep in $c$ from the first hidden

state

     **end for**

    **end for**

   **end for**

  **end for**

---

### 3.3. Case Description

This research conducted the environment test using Python based on the Gym library and the MPE library. By modifying the case environment accordingly, this study transformed the multi-agent problem into a single-agent problem for algorithm testing purposes.

Taking into account the real-world scenarios of drone delivery, we conducted performance evaluations based on standards set by Civil Aviation Administration of China (CAAC) and Federal Aviation Administration of the United States(FAA). These evaluations were performed using both a flight simulator called Airsim and actual flight data. Additionally, the study considered drone-specific flight characteristics, incorporated obstacle information, and had advance knowledge of customer coordinates for route planning [28].

Furthermore, the research investigated the impact of different numbers of drones on delivery efficiency and the environment, taking into account the collaboration between vehicles and drones. Below is an outline of the reinforcement learning environment configuration in this study relative to the case:

#### 3.3.1. Environment

This article employed Gym and MPE to construct the reinforcement learning environment. These tools are part of OpenAI's development toolkit for reinforcement learning

algorithms. They are compatible with various numerical computing libraries and neural network frameworks such as TensorFlow or PyTorch.

Gym is a widely-used library for creating general reinforcement learning environments, while MPE is a prominent framework for building and testing multi-agent environments. Within this environment, considering the parameters of both drones and vehicles, this study provided drones with a substantial action space to navigate rapidly toward their target points. Additionally, the environment accounted for obstacles and collision factors.

In terms of customer generation, the study took into account a customer distribution system modeled after Geographic Information Systems (GIS). Customers were distributed within individual stopping points, following a normal distribution based on their distance from the central point. This approach allows the multi-vehicle and drone delivery system to better align with real-world commercial models.

### 3.3.2. State

For different application problems, the definition of states may vary. Based on the characteristics of the truck–drone delivery model and this study, this article defines states differently for truck path planning and multi-drone aerial delivery.

In the context of truck path planning, discrete time slots are defined as states. The basic state represents the truck's movement between nodes, and a Monte Carlo tree is used as the exploration strategy for an individual vehicle. On the other hand, for multi-drone aerial delivery, continuous states are used. These states consist of drone coordinates, customer coordinates, and time. The composition of states is an important distinction. This study treats entire stopping points as basic state categories, utilizing Euclidean coordinates for the positions of trucks, drones, and customers. These coordinates are included as part of the state to facilitate the calculation of distances between points.

$$S_V = (P_i, r, t)$$

The number of drones is a crucial parameter in the truck–drone delivery problem. This study treats the number of drones as an adjustable parameter, and it tests and compares the impact of different drone quantities in various scenarios. Additionally, the MAPPO algorithm is applied to achieve more stable results in drone flights. A state $S$ is defined as

$$S_U = ((x_{c1}, y_{c1}), (x_{c2}, y_{c2}), \ldots, (x_C, y_C), (x_{d1}, y_{d1}), (x_{d2}, y_{d2}), \ldots, (x_D, y_D), r, t)$$

The definition provides the components of a state used in the environment, where $(x_c, y_c)$ represents the coordinates of customer locations; $C$ is the total number of customers in the mathematical model and environment parameters; $(x_d, y_d)$ represents the coordinates of other drone locations; $D$ is the total number of drones in the environment; $s$ represents the score; $t$ is the time.

With this state representation, the agents (vehicles or drones) are aware of the coordinates of other agents, which allows them to avoid collisions while efficiently reaching customer locations. This state representation, which includes customer positions and the positions of delivery vehicles, aligns more closely with real-world delivery scenarios.

### 3.3.3. Action

The action of the truck agent is to decide which parking point to move towards. The action set is as follows:

$$A_V = \{P_1, P_2, \ldots, P_p\}$$

For the drones, we utilize the Multi-Agent Particle Environment (MPE) framework, in which the flight trajectories of multiple drones are treated as continuous actions. Additionally, the potential collisions between drones and obstacles, as well as collisions between drones themselves, are considered as part of the action outcomes. The action set for each drone's agent is as follows:

$$A_U = \{Froward, Backward, Left, Right\}$$

These two action systems are tailored to address the specific requirements of truck and drone deliveries within the study's environment.

### 3.3.4. Reward

In this study, the reward system is structured as follows:

Positive Rewards: Positive rewards are granted to drones upon successful completion of delivery tasks. These rewards are determined by a base value multiplied by a coefficient generated based on the number of serviced customers in the area, following a normal distribution. Importantly, adhering to fundamental delivery principles, only the first drone to reach a customer's location is eligible for the reward. To mitigate challenges related to sparse rewards, which could lead to agents avoiding exploration, drones receive small rewards each time they approach a customer's coordinates during their flight trajectory. This mechanism encourages drones to complete deliveries more rapidly.

Negative Rewards: Negative rewards are introduced for specific scenarios, including redundant deliveries by drones, flight collisions, and contact with obstacles. These negative rewards are designed to discourage drones from getting entangled in these situations. The study dynamically adjusts and fine-tunes the reward settings during the learning process. As shown in Figure 4a, in the early experimental stages, collision penalties were initially set too high, inadvertently causing drones to prioritize avoiding each other. Consequently, the early reward curves predominantly yielded negative values.

Reward Evaluation: Considering the truck–drone cooperative delivery problem as a fully cooperative game from a game theory perspective, a Critic network is employed to evaluate all actions and centralize the handling of rewards. This enables the study to assess the contributions of individual drones to the delivery process. The study further normalizes each reward signal by subtracting the mean and dividing by the standard deviation. This normalization process aligns the reward signal distribution with a standard normal distribution, characterized by a mean of zero and a variance of one. The reward normalization strategy serves to expedite the learning and convergence process. In multi-agent environments, reward signals typically exhibit significant volatility, which can lead to training instability or divergence. Reward normalization helps mitigate this volatility, thereby enhancing training stability.

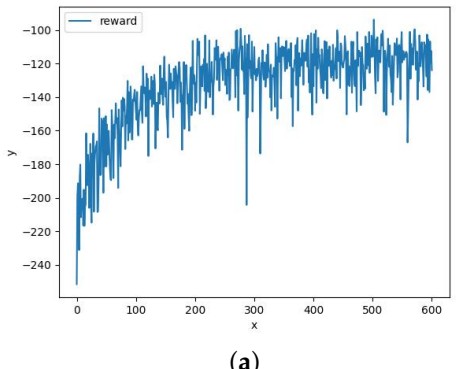
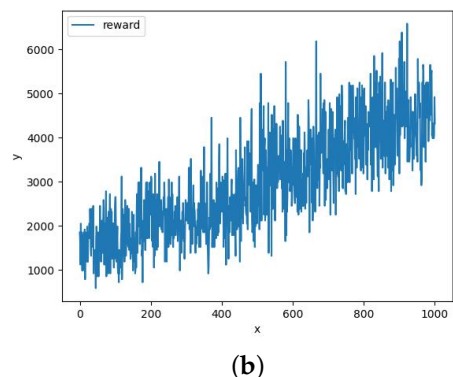

(a)                                                    (b)

**Figure 4.** MAPPO reward convergence curve. (**a**) MAPPO-Drone collision penalty is high; (**b**) MAPPO-After collision penalty adjustment.

## 4. Experiment and Results

In this section, we conducted testing on the Gym environment that we constructed. Firstly, we examined the learning efficiency and outcomes with and without action constraints. Secondly, we performed multiple tests within this model to assess the impact of different parameters on the results. Finally, we evaluated the effectiveness of the envi-

ronment and algorithm used, comparing it with the MAPPO algorithm under different parameter settings to assess the algorithm's capabilities.

All experiments were conducted using a consistent environment setup. The hardware environment included a CPU AMD R7-5800H-3.2 GHz, 16 GB DDR4 3200 MHz RAM, and an RTX3060 8 GB graphics card. The software environment consisted of Python 3.7.12, Pytorch 1.5.0, and Gym 0.10.5.

This study focused on the truck–drone transportation problem using the MPE framework. We conducted research to investigate the impact of collision penalties, different drone quantities, and varying customer quantities. We used a baseline scenario with two trucks working collaboratively and explored scenarios with a small number of drones (fewer than 5), a medium number of drones (5–12), and a large number of drones (more than 12). Customer quantities were distributed based on the number of drones, taking into account drone endurance. The minimum customer quantity was set to be equal to the number of drones, and the maximum customer quantity was set to three times the number of drones to simulate real-world conditions more effectively.

In terms of the environment setup, we treated the truck's path as a discrete problem, while the drone's flight trajectory problem was treated as a continuous one. This setting effectively simplified the action space, enabling solutions to the truck–drone problem at different levels. In this case, two trucks departed from specific starting points, visited all stopping points, and reached the final node. The drone's environment was randomized, and the customer distribution was based on known information. Drones received penalties as feedback when flying too close to obstacles or other drones.

Regarding drone flight and configuration in this study, we considered the urban context of truck–drone delivery. Since many cities do not permit heavy trucks to traverse urban areas, this study only considered medium-sized trucks as drone carriers. The significant size difference between drones with various payload capacities was also taken into account. The study prototypes included a 2 kg payload drone, currently used for experimental food delivery, and a 20-kg payload drone, utilized by some maritime resuply service providers for rapid ship deliveries. But, we are not looking at a combination of multiple drones right now. In this context, the minimum number of drones in our experiments was set to 4, and the maximum was set to 18.

Under the algorithm and environment settings of this study, it was inevitable for early drones to approach each other. Therefore, we reduced the penalty for drones getting too close to one another and increased the reward for drones reaching delivery points. The following image displays reward curves obtained using the MAPPO algorithm under various parameter configurations:

As shown in Figure 4, adjusting the penalty for drones' proximity helps achieve better rewards for reaching the target points. The main difference between Figure 4a,b lies in whether the penalty for drones getting close to each other is reduced. In Figure 3b, under an environment with 15 drones, 45 customers, and 6 obstructive buildings, the MAPPO algorithm ran for 5 million steps. Ultimately, it was found that the MAPPO algorithm can effectively approach the optimal solution step by step (with the highest reward reaching around 8000 after initial penalties were reduced).

By examining the reward curves for each game, the comparison in the above figure clearly shows that the MAPPO algorithm converges rapidly in small-scale drone testing. Although it appears challenging for MAPPO to converge in large-scale drone testing, this is due to the exponential increase in the volume of data that agents need to handle in a large-scale drone environment, resulting in slower learning efficiency.

From Table 2 and the reward curves above, it can be observed that MAPPO demonstrates a certain advantage when compared with other algorithms. Comparing with the MADDPG algorithm and the improved Twin Delayed MADDPG (MATD3) algorithm on top of MADDPG, it is evident that in small-scale testing, both MADDPG and MATD3 converge more slowly than the MAPPO algorithm, and their reward values are also lower.

**Table 2.** Algorithm result comparison.

| Algorithm-SCALE | Number of Drones | Number of Customers | Number of Obstacles | Number of Episodes | Number of Steps in a Episode | Average Reward |
|---|---|---|---|---|---|---|
| MAPPO-L | 15 | 45 | 6 | 100 | 1000 | 1372.61 |
| MAPPO-M | 8 | 12 | 4 | 100 | 200 | 173.24 |
| MAPPO-S | 4 | 6 | 3 | 100 | 100 | 68.96 |
| MATD3-L | 15 | 45 | 6 | 100 | 1000 | 1134.82 |
| MATD3-M | 8 | 12 | 4 | 100 | 200 | 168.19 |
| MATD3-S | 4 | 6 | 3 | 100 | 100 | 56.83 |
| MADDPG-L | 15 | 45 | 6 | 100 | 1000 | 965.71 |
| MADDPG-M | 8 | 12 | 4 | 100 | 200 | 121.53 |
| MADDPG-S | 4 | 6 | 3 | 100 | 100 | 37.52 |

Scale: L = Large, M = Medium, and S = Small.

From Figure 5, it is evident that a set of box plots, derived from randomly selected single trial results, showcases the superior performance of the MAPPO algorithm when compared to the MATD3 algorithm and the MADDPG algorithm. Even in the case of the medium-scale tests depicted in Figure 5b, while the mean performance of the MAPPO algorithm slightly trails behind that of MATD3, it surprisingly outperforms MATD3 in terms of the highest achieved performance values. Notably, MAPPO exhibits a considerably more stable result distribution when compared to the other two algorithms.

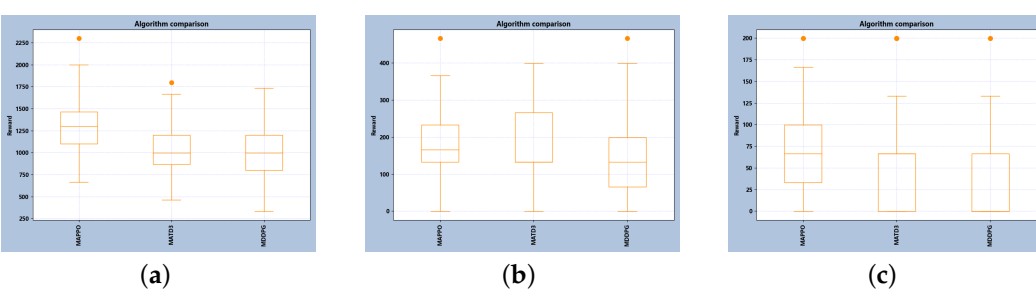

|       (a)       |       (b)       |       (c)       |

**Figure 5.** Result of a single run. (**a**) Test-L; (**b**) Test-M; (**c**) Test-S.

Specifically, as shown in Figure 6, from the reward curves, it is apparent that the MAPPO algorithm exhibits a faster and more stable upward trend in the environment of this study. The algorithm's computational efficiency also confirms this, as MAPPO shows a smoother increase, whereas the MADDPG algorithm not only takes more time but also exhibits greater reward curve fluctuations compared to the MAPPO algorithm. Furthermore, the MADDPG algorithm consumes more memory space than MAPPO.

Therefore, it can be concluded that the MAPPO algorithm holds advantages in terms of time efficiency, space efficiency, and results when solving the multi-agent truck–drone delivery problem. MAPPO can help achieve better results in the multi-agent environment of truck–drone delivery.

In summary, the experimental results in this case demonstrate that using the MAPPO algorithm for vehicle–drone intelligent delivery offers significant advantages. Furthermore, adjusting parameters and employing techniques such as reward normalization can further enhance the algorithm's learning efficiency and performance. Multiple experiments were conducted with varying numbers of agents and different parameters. These experimental findings hold valuable insights for the future development and optimization of vehicle–drone intelligent delivery systems [29].

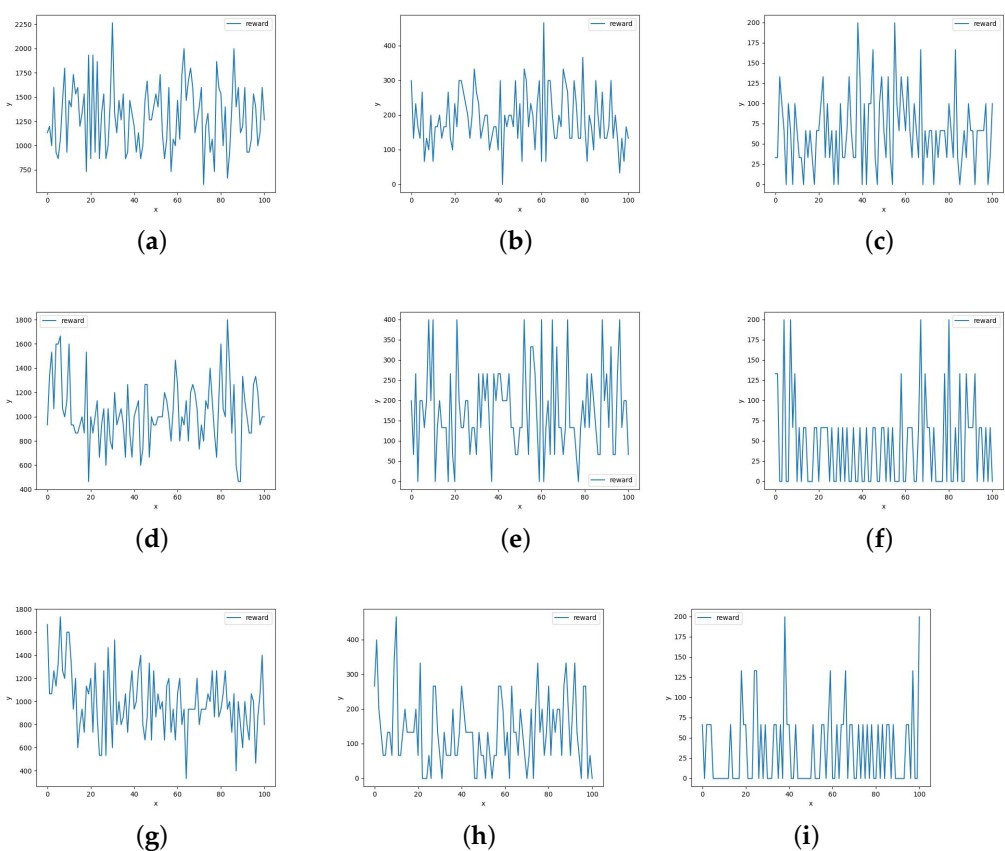

**Figure 6.** Algorithm reward graph. (**a**) MAPPO-Large case; (**b**) MAPPO-Medium case; (**c**) MAPPO-Small case; (**d**) MTD3-Large case; (**e**) MTD3-Medium case; (**f**) MTD3-Small case; (**g**) MADDPG-Large case; (**h**) MADDPG-Medium case; (**i**) MADDPG-Small case.

## 5. Conclusions and Future Research

In this paper, we addressed the truck–drone delivery optimization problem with multi-agent reinforcement learning. We employed the MPE framework and the MAPPO algorithm to tackle this complex problem, considering both truck routing and drone flight trajectories with multiple objectives. Leveraging the characteristics of truck–drone delivery, we adopted a hierarchical approach, utilizing Monte Carlo Tree Search (MCTS) and reward normalization techniques to accelerate training and enhance result convergence. This approach not only improved the speed of solving the problem but also expanded the methods for addressing vehicle–drone delivery.

In particular, in the fields of multi-agent systems and drones, we obtained significant insights through comparative experiments. We analyzed the performance of the MAPPO algorithm under different parameters, assessing its impact on training results and speed. After multiple experiments, we concluded that for practical applications of vehicle–drone delivery, it is advisable to minimize the number of customer points assigned to each drone, encouraging drones to consider only a single customer during flight [30]. Furthermore, we demonstrated that in the multi-agent environment of vehicle–drone delivery, the MAPPO algorithm outperforms the MADDPG algorithm in terms of time, space, convergence speed, and results. Thus, further research on various metrics of vehicle–drone delivery and the application of the multi-agent algorithm MAPPO to drone flight trajectories and service methods is valuable.

Regarding experiments, the use of reinforcement learning in the MPE environment allows for more versatile testing of different algorithms and simplifies comparative analysis and adjustments of reward functions. With the continuous technological advancements in e-commerce and logistics industries, the vehicle–drone delivery model becomes increasingly

research-worthy. Based on the research approach outlined in this paper, future research directions and improvements for vehicle–drone delivery models may include: Exploring the application of more multi-agent algorithms for testing and optimizing the reward function. Considering the use of advanced simulators like AirSim, based on game engines, to simulate real-world scenarios in 3D environments [10,31]. Conducting real-world experiments using actual drones for flight control or point-to-point delivery to expedite the application of drones in the logistics sector [32,33].

**Author Contributions:** Formal analysis, J.W.; Data curation, S.Q. and G.L.; Writing—original draft, Z.B. and X.G. All authors have read and agreed to the published version of the manuscript.

**Funding:** This research received no external funding.

**Institutional Review Board Statement:** Not applicable.

**Informed Consent Statement:** Not applicable.

**Data Availability Statement:** Data are contained within the article.

**Conflicts of Interest:** The authors declare no conflicts of interest.

## Abbreviations

The following abbreviations are used in this manuscript:

| | |
|---|---|
| UAV | Unmanned aerial vehicle |
| DQN | Deep Q-Network |
| MLP | Minimum latency problem |
| TRP | Traveling repair problem |
| TSP | Traveling Salesman Problem |
| TRPO | Trust Region Policy Optimization |
| MADDPG | Multi-Agent Deep Deterministic Policy Gradient |
| MAPPO | Multi-Agent Proximal Policy Optimization |
| MPE | Multi-Agent Particle Environment |

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
