# Peer review of "Truck-Drone Delivery Optimization Based on Multi-Agent Reinforcement Learning"

_drones, doi:10.3390/drones8010027_

Round 1

Reviewer 1 Report

Comments and Suggestions for Authors

The authors explore Enhancing Delivery Efficiency through Multi-Agent Reinforcement Learning in Truck-Drone Systems. Their study contributes to the existing body of knowledge by providing key findings. Overall, the study is well-written and insightful, making a valuable contribution to the field of UAV. The depth and clarity of your analysis are truly impressive. This work is a valuable resource for researchers and practitioners alike. With that said, there are still some details that require further discussion.

1.      I suggest a graphical representation for the introduction section.

2.      Summarize the existing body of literature on smarter deliveries, i.e., trucks and drones team up with AI in the form of a table. Also, mention here the weakness of previous works and the novelty of your proposed work.

3.      How do you handle communication and information sharing between agents to ensure coordinated decisions?

4.      How do you balance competing objectives within the reward function, delivery speed, cost efficiency, and energy consumption?

5.      What trade-offs are associated with different algorithms regarding training time, scalability, and performance?

6.      There are some sentence structure issues in the article, i.e.,

                                I.            the global logistics industry has been in need of, the global logistics industry has required a more efficient transportation model.

                             II.            Instead of “the partitioning of states and the adjustment of algorithm parameters”, author should write “the partitioning of states and adjusting algorithm parameters”.

                          III.            Instead of “in this study yields optimal solutions more readily compared to traditional heuristic algorithms”, author should use “in this study yields optimal solutions more readily than traditional heuristic algorithms”.

                          IV.            Instead of “Currently, traditional delivery models face challenges such as high labor costs, low efficiency, and the inability to rapidly deploy in urban environments”, use “Traditional delivery models face challenges such as high labor costs, low efficiency, and the inability to deploy in urban environments rapidly.”

                            V.            Instead of “Lastly, the adoption of truck-drone delivery models”, use “. Lastly, adopting truck-drone delivery models”.

                          VI.            Instead of “in particular, leverages the strengths of both trucks and drones.”, write “The truck-drone delivery model leverages the strengths of both trucks and drones.”

                       VII.            Instead of “Furthermore, compared to traditional vehicle delivery models, truck-drone joint delivery entails different tasks and requirements.” Write “Furthermore, truck-drone joint delivery entails different tasks and requirements compared to traditional vehicle delivery models.”

                    VIII.            Instead of “Considering the unique advantages of drones in achieving faster deliveries, this paper places a particular emphasis on the multi-agent problem, which is addressed through reward mechanisms and constraint conditions to better capture the complexities of real-world scenarios.” write “Considering the unique advantages of drones in achieving faster deliveries, this paper emphasizes the multi-agent problem, which is addressed through reward mechanisms and constraint conditions to capture the complexities of real-world scenarios better.”

                          IX.            Instead of “Moreover, this research takes advantage of third-party libraries,” write “Moreover, this research uses third-party libraries,”.

                            X.            Instead of “we must serve each and every customer,”, write “we must serve each/every customer,”.

                          XI.            Instead of “Regarding the objective function, this paper first defines a multi-objective problem”, write “This paper first defines a multi-objective problem based on time constraints regarding the objective function.”.

                       XII.            Instead of “we take into account the need for balanced completion”, write “we consider the need for balanced completion”.

                    XIII.            Instead of “The Actor component, which has its roots in policy gradient methods, is capable of selecting appropriate actions in continuous”, write “The Actor component, which has its roots in policy gradient methods, can select appropriate actions in continuous.”

Comments on the Quality of English Language

 Minor editing of the English language required

Author Response

Dear Editor

I would like to express my gratitude for the valuable feedback you provided on my paper. I truly appreciate your meticulous review and thoughtful guidance, which has positively influenced my research work. Building upon the modification suggestions you presented, I have carefully considered and revised the manuscript, providing detailed responses to each of your comments.

I believe these revisions not only enhance the quality of the paper but also align it more closely with the standards of academic writing. Once again, I appreciate your patient guidance. Should you have any further suggestions regarding my modifications or require additional information, please feel free to reach out.

Reviewer 2 Report

Comments and Suggestions for Authors

An algorithm based on reinforcement learning had been presented in the article for scheduling the optimal tour for a hybrid truck-drone delivery system. The details of the proposed mathematical model had been presented quite well in Section 3 of the article and the following are a few comments regarding certain parts of the model:

* The definition for the parameter d^(U)_{i,k} used in the mathematical notation (12) does not appear to be explicitly defined anywhere in the article, even though a reader may correctly guess its meaning.

* Does the model assume that more than 1 truck can occupy a certain parking point during the same time unit? If not, how is that reflected in the model and its constraints?

* Can you elaborate further on the reward model in this algorithm? It had been described in Section 4.3.3 to a certain extent, but is missing some details such as the mathematical model assumed for the reward process. Furthermore, the following statement in that section seems quite puzzling: "only the first drone to reach a customer's location is eligible for the reward". Does this assume that multiple drones will attempt to deliver the same item to the customer? If so, I am not aware of such a system existing, and it is more common to instead find only one drone attempting the delivery. The same question regarding the negative rewards which considers "redundant" deliveries.

The experimental results in Section 5 have demonstrated the application and merits of the proposed algorithm, and in comparison with other algorithms that attempt to achieve similar objectives. However, the section is missing the details of the experimental setup (e.g., the specifics of the reward model, and others) that makes it difficult for a reader to reproduce the same/similar results. Furthermore, it would be worthwhile to discuss/demonstrate the cost and/or complexity of the proposed algorithm relative to the other two algorithms (namely MADDPG and MATD) used in the analysis comparison. 

Author Response

(The authors gave the same response as above.)

Round 2

Reviewer 2 Report

Comments and Suggestions for Authors

Thank you for addressing the reviewers' comments and concerns in the revised manuscript.